# Development and Validation of an Inflammatory Prognostic Index to Predict Outcomes in Advanced/Metastatic Urothelial Cancer Patients Receiving Immune Checkpoint Inhibitors

**DOI:** 10.3390/cancers16081465

**Published:** 2024-04-11

**Authors:** Sara Mokbel, Giuilia Baciarello, Pernelle Lavaud, Aurelius Omlin, Fabio Calabrò, Richard Cathomas, Stefanie Aeppli, Pauline Parent, Patrizia Giannatempo, Kira-Lee Koster, Naara Appel, Philippe Gonnet, Gesuino Angius, Petros Tsantoulis, Hendrick-Tobias Arkenau, Carlo Cattrini, Carlo Messina, Jean Zeghondy, Cristina Morelli, Yohann Loriot, Vincenzo Formica, Anna Patrikidou

**Affiliations:** 1Faculty of Medicine, UCL—University College London, London WC1H 0AP, UK; sara.mokbel.17@ucl.ac.uk; 2Medical Oncology Department, Azienda Ospedaliera San Camillo Forlanini, 00152 Roma, Italy; gbaciarello@scamilloforlanini.rm.it (G.B.); gesuino.angius@uniroma1.it (G.A.); 3Medical Oncology Department, Gustave Roussy Cancer Campus, 94805 Villejuif, France; pernelle.lavaud@gustaveroussy.fr (P.L.); jean.zeghondy@gustaveroussy.fr (J.Z.); yohann.loriot@gustaveroussy.fr (Y.L.); 4Medical Oncology and Haematology Department, OnkoZentrum Zürich, 8038 Zurich, Switzerland; aurelius.omlin@ozh.ch; 5Medical Oncology 1, IRCCS National Cancer Institute Regina Elena, 00144 Rome, Italy; fabio.calabro@ifo.it (F.C.);; 6Department of Medical Oncology and Haematology, Cantonal Hospital St.Gallen, 9000 St. Gallen, Switzerland; stefanie.aeppli@kssg.ch (S.A.); kira-lee.koster@kssg.ch (K.-L.K.); 7Medical Oncology Departement, CHU Lille—Centre Hospitalier Régional Universitaire de Lille, 59000 Lille, France; pauline.parent@chu-lille.fr; 8Medical Oncology Department, Fondazione IRCCS—Istituto Nazionale dei Tumori, 20133 Milan, Italy; patrizia.giannatempo@istitutotumori.mi.it; 9Medical Oncology Departement, HUG—Hopitaux Universitaires Geneve, 1205 Geneva, Switzerland; naara.appel@etu.unige.ch (N.A.); philippe.gonnet@hcuge.ch (P.G.);; 10SCRI, London W1G 6AD, UK; h.arkenau@ucl.ac.uk; 11Maggiore della Carità University Hospital, 28100 Novara, Italy; carlo.cattrini@maggioreosp.novara.it; 12Ospedale A.R.N.A.S Civico, 90127 Palermo, Italy; carlo.messina@arnascivico.it; 13Medical Oncology Unit, Policlinico Tor Vergata, 00133 Rome, Italy; cristina.morelli@ptvonline.it (C.M.); vincenzo.formica@uniroma2.it (V.F.)

**Keywords:** immune checkpoint inhibitors, advanced/metastatic urothelial cancer, inflammatory markers

## Abstract

**Simple Summary:**

Immune checkpoint inhibitors (ICIs) are standard treatment for advanced/metastatic urothelial cancer (a/mUC) following progression on platinum agents. Traditionally, PDL-1 expression has been utilised to predict response to ICIs; however, this is not a robust selection marker. Therefore, alternative prognostic markers are needed to better select a subset of patients with a/mUC who are more likely to benefit. This study found that baseline systemic inflammatory profile and the absence of early serum inflammatory biomarker changes are associated with significantly better outcomes in patients. We developed and validated an immune prognostic index (U-IPI) specific to a/mUC patients on ICI treatment created using the five most statistically significant inflammatory markers. The U-IPI developed is an easily implementable prognostic tool to early predict benefit from ICIs.

**Abstract:**

Background: Immune checkpoint inhibitors (ICIs) improve overall survival (OS) in advanced/metastatic urothelial cancer (a/mUC) patients. Preliminary evidence suggests a prognostic role of inflammatory biomarkers in this setting. We aimed to develop a disease-specific prognostic inflammatory index for a/mUC patients on ICIs. Methods: Fifteen variables were retrospectively correlated with OS and progression-free survival (PFS) in a development (D, n = 264) and a validation (V, n = 132) cohort of platinum-pretreated a/mUC pts receiving ICIs at L2 or further line. A nomogram and inflammatory prognostic index (U-IPI) were developed. The index was also tested in a control cohort of patients treated with chemotherapy only (C, n = 114). Results: The strongest predictors of OS were baseline platelet/lymphocyte (PLR) and neutrophil/lymphocyte (NLR) ratios, and lactate dehydrogenase (LDH), NLR, and albumin changes at 4 weeks. These were used to build the U-IPI, which can distinctly classify patients into good or poor response groups. The nomogram scoring is significant for PFS and OS (*p* < 0.001 in the D, V, and combined cohorts) for the immunotherapy (IO) cohort, but not for the control cohort. Conclusions: The lack of a baseline systemic inflammatory profile and the absence of early serum inflammatory biomarker changes are associated with significantly better outcomes on ICIs in a/mUC pts. The U-IPI is an easily applicable dynamic prognostic tool for PFS and OS, allowing for the early identification of a sub-group with dismal outcomes that would not benefit from ICIs, while distinguishing another that draws an important benefit.

## 1. Introduction

Urothelial cancer, also known as transitional cell carcinoma, is the commonest histological type of bladder cancer, comprising approximately 90% of all cases [1]. The traditional first-line (L1) treatment for advanced/metastatic urothelial cancer (a/mUC) is a platinum-based combination chemotherapy regimen, very recently replaced (at least in terms of strength of evidence) by the combination of enfortumab vedotin/pembrolizumab [2,3], although this is not yet approved in Europe. Traditional outcomes were unsatisfactory, as the median overall survival (mOS) is approximately 15 months in patients receiving a cisplatin-based regimen, with median progression-free survival (mPFS) at 7–8 months [4,5,6]. Those receiving a carboplatin regimen due to being ‘unfit’ for cisplatin have inferior outcomes, with an mOS of around 9 months and mPFS of 5–6 months [7]. The addition of switch maintenance ICI avelumab has increased the mOS to 21.4 months [8] but is applicable only for patients who have not progressed under L1 platinum-based chemotherapy, therefore excluding the patients with very poor prognosis. ICIs are also available as L1 therapy for patients who are cisplatin ‘unfit’ and PD-L1-positive and those ineligible for any platinum treatment regardless of PD-L1 status [9,10].

ICIs, notably pembrolizumab, is the standard second line (L2) therapy for patients with platinum-refractory a/mUC who have not received avelumab [10,11]. However, they are not successful in improving outcomes in all comers. PD-L1 expression is not a robust biomarker for selecting patients who will respond well to ICIs. This is due to a lack of reliable reproducibility owing to its dynamic expression and the lack of harmonisation of PD-L1 assay testing which makes interpretation of the results challenging [12].

Alternative prognostic markers have been researched to predict the efficacy of ICIs in patients with a/mUC, although none have been fully validated for routine use clinically. Biomarkers that have been shown to be helpful prognostic indicators of response include tumour mutational burden (TMB) [12], CD8 T-cell expression [13], and interferon-gamma gene signature [13]. Additionally, new potential biomarkers are emerging, such as circulating tumour DNA and microbiota [14,15]. It has been shown that high ctDNA levels correlates with more aggressive and advanced forms of disease and is independently prognostic for overall survival in patients treated with durvalumab [16]. Furthermore, the mismatch repair (MMR) status in a/mUC might have a prognostic role; however, the low number of bladder cancers with such alterations limits its use in practice [17].

One of the most informative tissue-based biomarkers is the recently described gene expression Immunotherapy Response Score (IRS), which integrates TMB and the expression of the PD-1, PD-L1, TOP2A, and ADAM12 genes [18]. IRS has been assessed as informative in a post hoc analysis of the phase 2 IMvigor210 trial, with an mOS of 16.46 months vs. 7.43 months for the IRS-high vs. IRS-low patients, respectively (*p* < 0.001) [19].

Given that immune inflammatory cells can be actively tumour promoting [20], systemic inflammatory markers have been investigated for their role in predicting response vs. resistance to ICIs. The derived neutrophil–lymphocyte ratio (dNLR) and lactate dehydrogenanse (LDH) have proven to be valuable in predicting response to ICIs. The two largest retrospective studies on advanced melanoma patients treated with ipilimumab demonstrated the independent prognostic value of a dNLR ≥ 3 and LDH of at least 2.5 times the upper limit of normal (ULN). Both studies found that higher pro-inflammatory markers predicted poorer outcomes to treatment [21,22]. The prognostic utility of the NLR has proven to be universally applicable across different cancer types, including genitourinary cancer [23].

Combining the value of these strong prognostic markers, baseline dNLR and LDH were used to develop a tool to predict outcomes in advanced NSCLC patients receiving ICI therapy [24,25,26]. The lung immune prognostic index (LIPI) score categorised patients into 3 risk groups (poor, intermediate, and good) with significant differences identified between the groups with regard to PFS and OS in the ICI cohort but not for the chemotherapy cohort [24]. LIPI has been shown to be prognostic in other solid tumours treated with ICIs such as renal cell carcinoma (RCC) and melanoma [27].

Some evidence exists regarding the use of inflammatory markers in predicting response to ICIs in a/mUC. The LIPI score seems to be promising in accurately predicting outcomes in this cohort. However, further prospective studies need to be conducted to further validate its use; furthermore, although prognostic, it is not exclusively predictive for ICI treatment [28]. It therefore remains challenging to identify patients with a/mUC who would specifically benefit from ICIs. This selection is paramount to improve therapeutic outcomes for patients, avoid unnecessary adverse reactions to treatment, and allow alternative, more suitable treatment options to be selected earlier.

The aim of this study was to develop and validate an immune prognostic index (U-IPI) specific to a/mUC on ICI treatment.

## 2. Materials and Methods

Two independent cohorts of patients were retrospectively analysed across ten European centres. The main cohort (IO) included patients treated from May 2012 to May 2022 with histologically confirmed platinum-refractory a/mUC who were subsequently treated with ICI at L2 or further. Patients receiving avelumab as switch maintenance after lack of progression under L1 chemotherapy were excluded. Within this cohort, patients were assigned to a development (D) sub-cohort, used to create the U-IPI index, and a validation (V) cohort, to independently validate the developed index.

A control cohort (C) of patients who had never received ICIs was also used, for which the L1 treatment of platinum-based chemotherapy was used as a reference timepoint.

The primary endpoints were OS and PFS. OS was defined as the time between ICI start date and death from any cause. PFS was defined as the time period between treatment start date and disease progression or death. Patients not reaching the respective endpoints were censored at the last known follow-up date.

Baseline characteristics such as age at treatment start date, gender, anatomic location and number of metastatic sites, line of IO, and ECOG status were collected and analysed for each participant. Serum inflammatory marker variables were collected at baseline (within one week prior to starting ICI treatment), and at 4 weeks post (±1 week) ICI treatment start. These variables were neutrophil, lymphocyte, and platelet count, neutrophil-to-lymphocyte ratio (NLR), platelet-to-lymphocyte ratio (PLR), lactate dehydrogenase (LDH), and albumin levels. Missing values were imputed. All patients gave informed consent to participate in the clinical trials, and all data collected were anonymised and coded.

Univariate and multivariate (MVA) analyses, as well as LASSO analysis, were used for the identification of significant variables and the construction of the U-IPI index. Kaplan–Meier curves were built for PFS and OS for each cohort and sub-cohorts. To compare different survival curves, the log rank test was used. Significance for inclusion of variables in the MVA was set at *p* < 0.05. Significance for the U-IPI index prognostic ability was set at *p* < 0.001). Statistical analysis was performed with the SPSS software package (v.25, IBM Corp, Armonk, NY, USA, 2017).

## 3. Results

A total of 510 patients were included in this analysis; 396 patients were included in the IO cohort, 264 assigned to the D cohort, and 132 assigned to the V cohort. The median length of follow-up was 41 months. Median OS was similar in the D, V, and combined cohorts (15.3 months vs. 14.2 months vs. 14.6 months, *p* = 0.889). A total of 114 patients were analysed in the control (chemotherapy) cohort (mOS: 14.3 months). Baseline patient clinical–pathological characteristics are summarised in Table 1. No patient received recently approved life-prolonging therapies such as enfortumab vedotin, erdafitinib, or sacituzumab govitecan.

### 3.1. Development Cohort

In the D cohort, a total of 22 variables were analysed in the UVA. Of these, 15 variables were included in the MVA, and were optimally dichotomised with maximally selected rank statistics. Using MVA and LASSO analysis, five variables were retained as most significant (*p* < 0.05) and were used for the development of the U-IPI index (baseline NLR, baseline PLR, neutrophil change at 4 weeks, albumin change at 4 weeks, LDH change at 4 weeks). These variables were used to build the U-IPI nomogram for the prediction of survival outcomes for patients. Individual component scores are added to obtain the final score, which ranges from 0 to 413. The nomogram is able to predict mOS and survival probabilities at 3 and 6 months for platinum-refractory a/mUC patients receiving ICI treatment (Figure 1).

The best total points cut-off was defined at 170 with the use of maximally selected rank statistics, separating the development cohort into good-risk (<170 points) and poor-risk (>170 points) U-IPI categories (Figure 2A), with an mOS of 27.6 months vs. 5.2 months, respectively (*p* < 0.0001, HR for the good-risk category: 0.1833; 95% CI: 0.1226; 0.2739).

### 3.2. Validation Cohort

The application of the developed U-IPI index in the V cohort resulted in significantly different mOS in the two risk categories (26.6 months vs. 5.5 months, *p* = 0.0005, HR for the good-risk category: 0.4299; 95% CI: 0.2682; 0.6891, Figure 2B).

### 3.3. Combined IO Cohort

The developed U-IPI index was also applied to the combined IO cohort (n = 396), with 244 patients (61.6%) attributed to the good-risk category and 152 patients (38.4%) to the poor-risk category. It retained significance in terms of mOS (26.2 months vs. 5.4 months, *p* < 0.0001, HR for the good-risk category: 0.2767; 95% CI: 0.2048; 0.3737, Figure 2C).

The index was also prognostic when applied for PFS, with mPFS of 6.1 months, 2.2 months, and 3.7 months for the good-risk, poor-risk, and combined cohort, respectively (*p* < 0.0001, HR for the good-risk category: 0.3972; 95% CI: 0.3090; 0.5107, Figure 2D).

### 3.4. Control (Chemotherapy) Cohort

The application of the U-IPI in the C cohort resulted in non-significant separation for mOS (16.3 months vs. 11.4 months vs. 14.3 months for the good-risk, poor-risk, and overall cohort, *p* = 0.0406, Figure 3A), confirming its specificity for ICI therapy.

The index was not predictive for mPFS in the C cohort either (7.7 months, 5.8 months and 7.0 months for the good-risk, poor-risk, and overall cohort, *p* = 0.2022, Figure 3B).

## 4. Discussion

Despite the demonstrated effectiveness of ICI in a/mUC, only a small proportion of those receiving treatment will have a clear objective benefit, whilst a larger number will be exposed to potentially serious toxicities with no improvement in quality of life or survival. Several trials have demonstrated that the proportion of platinum-refractory a/mUC showing objective response to ICIs range from 13 to 21%, yet these patients are mainly long-term responders [11,12]. Hence, the issue of poor response should not tarnish the efficacy of ICIs as a whole in this population. However, it reinforces that the success of ICIs in this demographic is highly reliant on accurately selecting a subset of patients who will benefit best prior to or very early on upon treatment initiation.

The U-IPI score described in this study can very distinctly classify between response groups. In the development cohort, patients with a good U-IPI score (<170 points) have a significantly higher median mOS of 30.5 months compared to 5.0 months for those with a poor score (>170 points). The model’s success was reproducible when applied to the combined cohort for PFS and OS. In fact, the index is able to identify, based upon combined information from the baseline profile and early changes in serum inflammatory markers, a group of good prognosis patients whose mOS at 26.2 months is significantly higher that this for permbrolizumab in the second line (mOS = 10.2) [10,11] and indeed comparable to more recently reported first-line combination regimens.

The U-IPI was formulated with the five variables, selected after LASSO regression analysis, deemed to be the most significant prognostically for mOS: baseline PLR, baseline NLR, LDH changes at 4 weeks, albumin changes at 4 weeks, and NLR change at 4 weeks. This method of analysis involves performing both selection and regularisation, thus increasing the accuracy of the U-IPI produced. The combination of multiple variables into a single prognostic tool instead of using each separately increases the index potency.

The lower the U-IPI score, the better survival outcomes were for this study cohort. A lower score clinically translates to less thrombocytosis, less lymphopenia, and a lack of neutrophilia. This therefore suggests a lower baseline systemic inflammatory profile and a reduced acute inflammatory response (reflected by changes in inflammatory markers at 4 weeks) correspond with better responses to ICIs. These results are consistent with previous studies looking into the predictive impact of changes in inflammatory biomarkers in patients with advanced cancers treated with IO. A retrospective study of 90 patients in phase 1 trials showed baseline and early changes at 6 weeks in NLR and PLR values were strongly associated with clinical outcomes. Higher values were associated significantly with worse OS and PFS (*p* < 0.05) [29]. Similarly, a prospective study of 313 patients with metastatic renal cancer treated with IO evaluated the association of the systemic immune inflammation index (SII: platelets × neutrophils/lymphocytes), NLR and PLR with survival. The study showed that higher baseline SII, PLR, and NLR values were able to predict outcomes, and that SII changes at 3 months predicted OS (*p* < 0.0001) [30]. Although these studies are not specific to a/mUC, their shared findings strongly support our results.

When compared to the two most important inflammatory prognostic models existing for use in a/mUC, U-IPI seems to have superior prognostic powers, but also a predictive ability (specificity for response to ICIs). LIPI, the strongest inflammatory prognostic index across oncology, was shown to be informative for survival in a real-world retrospective cohort, as well as in a validation cohort consisting of trial patients, with an impressive HR of 2.69 and 2.89, respectively. Indeed, when the LIPI is applied in a/mUC, it is able to identify a good-prognosis cohort with an mOS of 19.7 months vs. 14.4 months for the overall cohort [28]. It was not shown to be superior to the Bellmunt score, a stratification factor for a/mUC patients receiving chemotherapy having progressed on platinum-based chemotherapy [31]. U-IPI, while having a very comparable mOS for the combined IO cohort (14.6 months), has the discriminatory ability to identify very early a good-risk group with an impressively better survival, prolonged by an entire year compared to the combined cohort (mOS: 26.2 months). The poor prognosis patients per U-IPI seem to have an identical prognosis with the poor prognosis patients identified by LPI (mOS of 5.4 mo for both). Furthermore, LIPI was also prognostic for patients treated by chemotherapy, with longer survival for the good prognosis patients, whilst U-IPI seems to be specific for ICI treatment.

A more recently described index, described on a/mUC patient cohort exclusively receiving second-line atezolizumab, included seven risk factors, and distinguishes four risk categories [32]. It identifies a good-prognosis cohort of 18.6 months, slightly lower than LIPI and U-IPI, and refines and even worse prognostic group at 2.1 months for mOS. However, it is described for patients treated with an ICI that is rarely used nowadays for m/aUC patients.

A major advantage of the U-IPI in the direction of its potential clinical use is that the variables constituting the index are collected routinely as part of standard practice when using IO, and its predictive ability is clinically meaningful, providing information that could enable a binary yes/no decision in terms of ICI use in specific treatment, given the important difference in OS of the two prognostic groups. Even when compared to the IRS, which is based on more sophisticated gene-expression techniques, U-IPI appears to be a more powerful classifier in its discriminant ability regarding survival outcomes on ICIs, given the greatest separation of the survival curves between high- and low-score groups. Therefore, a large portion of patients with a/mUC can benefit from its use to help guide treatment options to optimise survival outcomes. Additionally, the U-IPI combines variables taken at baseline and at 4 weeks into treatment, which allows for very early identification of poor versus good responders to ICIs. This means that treatment is not restricted or withheld even if baseline variables are not favourable and as they may subsequently receive a good U-IPI score after all five variables are considered. If patients are deemed unlikely to receive benefit from ICIs after U-IPI analysis, treatment can be terminated early on to reduce the potential unnecessary toxicities and costs associated with ICI use.

The U-IPI’s prognostic scoring ability, despite being based on variable association with OS, is statistically significant (*p* < 0.0001) for both for OS and the secondary endpoint PFS. The nomogram scoring identified a subset of patient who respond extremely well to ICI treatment in a previously platinum resistant setting. Further adaptation of the U-IPI might identify an ‘intermediate’ repose group, seen in other prognostic tools such as LIPI. This allows for a group where treatment may still improve survival outcomes, but further assessment would be warranted on a case-by-case basis.

This study design inherently has several limitations, notably selection bias owing to the retrospective nature and the relatively small validation cohort. Furthermore, given that the control cohort of patients that have never received immunotherapy corresponds to patients treated over a decade ago in their majority (pre-immunotherapy era), this led to a large number of missing data, such as ECOG PS status and radiological response for the chemotherapy cohort, which correspond to potential significant source of bias in the OS and PFS assessment, respectively. Additionally, this design is subject to missing data that were not accounted for at baseline and at 4 weeks.

To support the results found, further studies into characteristics that constitute good responders in ICI groups compared to chemotherapy groups alone will further strengthen any prognostic markers identified.

## 5. Conclusions

We developed and validated the U-IPI, an inflammatory prognostic index for patients with a/mUC cancers receiving ICI therapy at the platinum-refractory setting. The lack of a baseline systemic inflammatory profile and the absence of early serum inflammatory biomarker changes are associated with better outcomes on ICIs in a/mUC pts. The U-IPI nomogram developed is a dynamic prognostic tool for PFS and OS, identifying early a sub-group that likely does not benefit from ICIs.

Prospective validation in larger independent clinical trials is required prior to clinical applicability in the aim of providing an accurate, individualised prognostic prediction on survival outcome in this patient group. As the nomogram combines serum markers routinely assessed in patients, it can be easily used by oncologists in their daily practice.

## Figures and Tables

**Figure 1 cancers-16-01465-f001:**
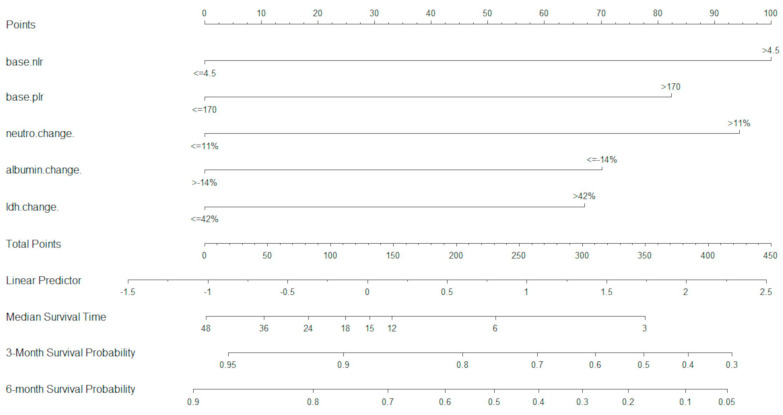
U-IPI nomogram for overall survival. Median overall survival is measured in months.

**Figure 2 cancers-16-01465-f002:**
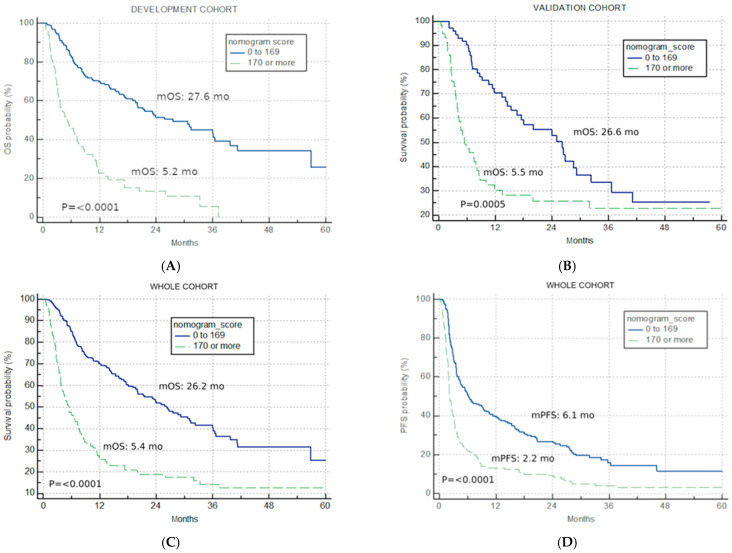
Univariate analysis. Kaplan–Meier curves for nomogram score cut-off (good risk < 170, poor risk > 170) (**A**) overall survival in the development cohort; (**B**) overall survival in the validation cohort; (**C**) overall survival in the combined IO cohort; (**D**) progression-free survival for the combined IO cohort.

**Figure 3 cancers-16-01465-f003:**
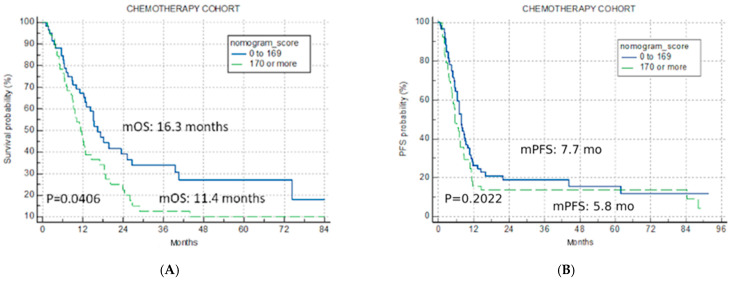
Univariate analysis chemotherapy cohort. Kaplan–Meier curves for nomogram score cut-off for two risk categories (good risk < 170, poor risk > 170) (**A**) overall survival chemotherapy cohort; (**B**) progression-free survival for chemotherapy cohort.

**Table 1 cancers-16-01465-t001:** Baseline patient characteristics.

Characteristics	ICI Cohort N = 424	Chemo N = 160
**Gender (n, %)**		
Male	337(79)	127 (79)
Female	87 (21)	33 (21)
**Age at treatment start** (years) (median, range)		
	68 (37–92)	-
**Smoking history (n, %)**		
Never	89 (21)	32 (20)
Current	199 (47)	75 (47)
Former	93 (22)	24 (15)
Unknown	43 (10)	29 (18)
**PD-L1 status (n, %)**		
Positive	46 (11)	NA
Negative	62 (15)	7 (4)
Unknown	316 (74)	153 (96)
**Primary tumour (n, %)**		
Upper tract (pyelocaliceal cavities, ureter)	75 (18)	34 (21)
Lower tract (bladder, urethra)	299 (70)	125 (78)
Mixed	5 (1)	1 (1)
Unknown	45 (11)	0
**Histology (n, %)**		
TCC	351 (83)	131 (82)
SCC	12 (3)	16 (10)
Other	13 (3)	8 (5)
Unknown	48 (11)	5 (3)
**Type of chemo treatment (n, %)**		
Platinum-based therapy	-	87 (54)
Gemcitabine only	-	1 (1)
Vinflunine only	-	9 (6)
Paclitaxel only	-	6 (4)
Other	-	57(35)
**Type of ICI treatment (n, %)**		
PD-L1 inhibitor	204 (48)	-
PD-1 inhibitor	220 (52)	-
**Number of metastatic sites (median, range)**	2 (0–6)	2 (0–24)
**IO line (median, range)**	2 (1–6)	-
**Pretreatment performance status (ECOG) (n, %)**		
0	169 (40)	33 (21)
1–2	242 (57)	42 (26)
≥3	8 (2)	1 (1)
Unknown	5 (1)	84 (52)
**Circulating inflammatory markers (median, range)**		
Baseline platelets	250 (32.3–796)	317 (52–617,000)
Baseline neutrophils	4.9 (1.0–27.6)	6.54 (1.09–15,110)
Baseline lymphocytes	1.3 (0.2–4.5)	1.7 (0.11–3290)
**Last known status (n, %)**		
Deceased	269 (63)	115 (72)
Alive	155 (37)	45 (28)
**Radiological response (n, %)**		
Complete response	38 (9)	10 (6)
Partial response	75 (18)	30 (19)
Stable disease	91 (21)	14 (9)
Progressive disease	193 (46)	17 (11)
Non evaluable	27 (6)	89 (55)
**Overall survival (Days) (median, range)**	260.5 (4–3162)	365 (28–2898)

## Data Availability

The datasets used and/or analysed during the current study are available from the corresponding author on reasonable request. Anna Patrikidou had full access to all the data in the study and takes responsibility for the integrity of the data and the accuracy of the data analysis.

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
