# Peer review of "Development and Validation of an Inflammatory Prognostic Index to Predict Outcomes in Advanced/Metastatic Urothelial Cancer Patients Receiving Immune Checkpoint Inhibitors"

_cancers, 2024, doi:10.3390/cancers16081465_

Round 1

Reviewer 1 Report

Comments and Suggestions for Authors

Title - concise and clear - No Remarks

Abstract - clearly defining the essence of the study - 

row 46 - IO cohort? - abbreviation needs clarification - MINOR

introduction - elegant presentation on the current knowledge on the subject, and the different approaches for ICI inclusion in therapy of a/m urothelial cancer 

row 70 PDL-1 ? typo? Minor

row 87 - IO ? - needs clarification

Material and Methods - No Remarks

Results 

row 141 - "Median OS was similar in the D, V and cohorts (15.3 months vs 14.2 months vs 14.6 months, p=0.889)" - third cohort is missing, what is this third cohort? - MAJOR   table 1 - primary tumor location - Upper Tract (Bladder) ???

Lower tract (ureter, renal pelvis)???? - MAJOR - serious discrepancy   Type of chemo treatment (n, %) - row misalignment - MAJOR

Pretreatment performance status (ECOG) (n,%) - extremely high rate of unknown in chemo group (52 %) - potential significant source of bias in OS assessment  - needs comment - Major

Circulating inflammatory markers (median, range)  - row misalignment - MAJOR Radiological response (n, %) -  extremely high rate of non-evaluable in chemo group (55%) - potential significant source of bias in PFS assessment  - needs comment - Major   Discussion - No remarks - nicely implementing the authors results into the contemporary literature and strongly substantiating their conclusions

Reviewer 2 Report

Comments and Suggestions for Authors

The manuscript "Development and validation of the U-IPI index to predict out- comes in advanced/metastatic urothelial cancer patients receiving immune checkpoint inhibitors" aims to develop and validate an immune prognostic index for  advanced/metastatic urothelial cancer. The manuscript brings novelty to urothelial cancer prognosis and treatment, due to the identification of an inflammatory profile able to stratify the patients.

In the abstract, provide the acronym for LDH.

The discussion should be improved. The authors could discuss more the results otained and the inflammatory parameters with the literature. 

Round 2

Reviewer 1 Report

Comments and Suggestions for Authors

the authors has taken into account this reviewer`s recommendations